# The Reliability and Accuracy of Palpation, Radiography, and Sonography for the Detection of Keel Bone Damage

**DOI:** 10.3390/ani9110894

**Published:** 2019-11-01

**Authors:** Linnea M. Tracy, S. Mieko Temple, Darin C. Bennett, Kim A. Sprayberry, Maja M. Makagon, Richard A. Blatchford

**Affiliations:** 1Department of Population Health, University of Georgia College of Veterinary Medicine, Athens, GA 30602, USA; Linnea.Tracy@uga.edu; 2Animal Science Department, College of Agriculture, Food and Environmental Sciences, California Polytechnic University, San Luis Obispo, CA 93407, USA; smtemple@calpoly.edu (S.M.T.); dbenne06@calpoly.edu (D.C.B.); kspraybe@calpoly.edu (K.A.S.); 3Center for Animal Welfare, Animal Science Department, College of Agriculture and Environmental Sciences, University of California, Davis, CA 95616, USA; mmakagon@ucdavis.edu

**Keywords:** laying hen, keel bone, palpation, radiography, sonography, reliability

## Abstract

**Simple Summary:**

Due to the recent increase in reports of the prevalence of keel bone damage in laying hens, this has become a topic of welfare concern. Keel bone damage is often in the form of a fracture, and therefore may compromise the hen’s welfare. Detecting keel bone damage in live hens has been problematic, as the bone must either be palpated, which is a measure poor in accuracy, or dissected, in which case the hen must be sacrificed. However, use of imaging technology is increasing in experimental studies. We set out to determine if training with feedback on accuracy could improve palpation accuracy, as well as to determine the accuracy of portable radiography and sonography to detect keel bone damage. Even with feedback, palpation remained an inaccurate method, while both radiography and sonography showed high accuracy for detecting fractures. These two techniques show promise in detecting keel bone fractures in live birds.

**Abstract:**

Palpation is the most popular method of measuring keel bone damage on live birds, although it has been criticized for being subjective and inaccurate. The goals of this study were to examine intra- and inter-rater reliability when trained with feedback of accuracy, as well as determine the accuracy of portable radiography and sonography. Four evaluators palpated 50 103-week old Lohmann LSL-lite hens immediately following euthanasia. Of those birds, 34 were then radiographed, sonographed, and all 50 were re-palpated. Lastly, the keels were dissected and scored. The presence of deviations (DEV), fractures (FR), and tip fractures (TFR) was scored for each method. Reliability of palpation was analyzed using Cronbach’s Alpha (intra) and Fleiss’ Kappa (inter) tests. Radiography and Sonography scores were further compared with dissection scores to determine sensitivity and specificity. Initial inter-observer reliability was 0.39 DEV, 0.53 FR, and 0.12 TFR, with similar scores for the second round of palpation. Scores for intra-observer reliability ranged from 0.58–0.79 DEV, 0.66–0.90 FR, and 0.37–0.87 TFR. A high prevalence of TFR, but low assessor agreement, warrants the development of specialized training for the palpation of this area. Both radiography and sonography showed relatively high sensitivity for FR and TFR, but low for DEV. On the other hand, specificity was generally high across all damage types. Even with feedback, palpation reliability was poor. However, portable radiography and sonography show promise for detecting keel fractures.

## 1. Introduction

The keel bone extends approximately 9–12 cm on the ventral sagittal plane of a laying hen. The modified sternum anchors flight muscles on a ventral spine that tapers to the caudal tip. The keel does not fully ossify until the early stages of the lay period (between 28 and 40 weeks) and the caudal tip often remains cartilaginous. This makes the bone more vulnerable to damage during the pullet move-in process as well as later in a laying hen’s life than other long bones which complete ossification earlier [1,2,3]. 

Measuring the prevalence of keel bone fractures is necessary to understanding the scope of the effect such fractures have on hen welfare and productivity and for validating the efficacy of preventative measures. However, assessing fracture prevalence and incidence at various life-stages using necropsy can prove laborious, expensive, and damaging to flocks in the long-term. Thus, palpation of the keel bone has been presented in the past [4,5,6] as a potentially viable, low-cost technique for detecting keel fractures in live laying hens. Currently, keel fracture prevalence is estimated between 23% and 96% of European laying hens depending on housing system and age [7,8], although little information is available for flocks in the United States. Fractures and deformations of the keel are often considered to decrease the welfare of affected birds [9,10,11] and decrease a hen’s laying potential [12,13]. As the international egg industries begin the switch from conventional cages to systems associated with higher keel fracture prevalence rates such as furnished cages and aviary systems, developing an understanding of risk factors and the efficacy of preventative measures will become essential to preserving layer hen welfare.

Previous studies have also indicated that the use of radiographic methodologies such as traditional radiography (X-rays), computed tomography (CT), magnetized resonance imaging (MRI), and sonography (ultrasound) are useful and often highly accurate in detecting keel damage [6,9,14,15]. However, these methods are often impractical to implement on a broad scale due to concerns about radiation safety (radiography and CT), accessibility, expense, time involvement, and interpretation. Sonography has been proposed as a valid mobile methodology for keel damage detection as it shares radiography’s advantage in rapidity without the associated radiation safety risk, although as yet it has not been quantitatively compared to radiography’s detection values. Both of these technologies are now available as fairly lightweight digital portable units, though they may pose a biosecurity risk if used on different commercial farms. Additionally, radiography and sonography may be useful in furthering the standardization and reliability of hand-palpation training.

In this study we examined the intra-observer reliability of a previously published keel bone palpation technique [4,5] and modified the technique by allowing the training palpators to examine the dissected keels in comparison to their previous palpation scores of the same birds. After the completion of training rounds, final palpation scoring was validated against the dissected keels of the palpated hens. Additionally, we examined the accuracy of radiography and sonography for the detection of such fractures and deviations, validated similarly against the subsequently dissected keel bones. Prior to our study, few publications have assessed the use of radiography or sonography in identifying keel bone fractures in laying hens [9,16], and none have compared the reliability of palpation with radiography, nor validated radiographic or sonographic results with gross dissection. 

## 2. Materials and Methods 

All procedures used in this study were approved by the California Polytechnic State University Animal Care and Use Committee (IACUC) Protocol #1613. 

### 2.1. Hen Housing

Lohmann LSL-lite hens (103 weeks of age and at end of cycle) were housed in a Salmet enriched colony system (SALMET AGK 3600 Enrichable Colony System, SALMET GmbH & Co. KG, Dietzenbach, Germany) at the Cal Poly Poultry Center at California Polytechnic State University. The house consisted of two stories, with 64 cages per floor (25 hens/cage). Each cage measured 360 × 63 × 56 cm (L × W × H) with a nest box, scratch pad and two levels of perching, a low perch 9 cm from the floor and a high perch 25 cm form the floor. Feed and water were provided ad libitum. Fifty cages were selected randomly throughout the house and one hen per cage was euthanized with carbon dioxide gas. Immediately following euthanasia, hens received a tape leg tag identifying them by cage number. All keel bone assessments were performed immediately following euthanasia.

### 2.2. Palpation

The four assessors consisted of two experienced palpators (based on training of over 100 live keel palpations in the field, [5]), one moderately experienced palpator, and one inexperienced palpator (no keel palpation experience). Before the start of palpation, the 4 assessors were trained on normal keel bone anatomy. To palpate, assessors were instructed to run their thumb and forefinger against the hen along the sagittal axis of the keel bone no more than twice to feel for the presence or absence of deviations/deformations (portions of the keel structure that deviate from a perfectly straight, 2-dimensional axial line, including palpable divots on the surface of the bone), fractures (evidenced by deposits along the surface or sides of the bone indicating a possible fracture callus, or palpable gaps in the bone), and keel bone tip fractures (approximately 2 cm of the most caudal aspect of the bone). Assessors performed the palpation of each hen before checking the hen’s identifying cage number tag and recorded the score each category. Hens were palpated once by each assessor, then radiographed and sonographed, and then palpapated a second time by each assessor. Assessors did not discuss their results, nor did they view the score sheets of their peers. 

### 2.3. Radiography and Sonography

Hens were radiographed by a licensed veterinarian using a portable MinXray HF 100/30 Ultralite, with images obtained using a Toshiba D-124S X-ray tube. To limit personnel exposure to radiation, only 34 hens were radiographed. Radiographs were scored by a veterinarian for the presence or absence of fractures, deviations, and tip fractures on the keel bone. 

The same 34 hens were then sonographed by trained personnel using a SonoSite Edge II, with images obtained using a 5–8 MHz transducer. The probe was passed along the ventral aspect of the keel in the sagittal plane, and the resulting image was scored as indicating the presence or absence of fractures, deviations, and tip fractures. Hens had not entered rigor mortis at the time of palpation, radiography, or sonography.

### 2.4. Dissected Keel Bone Scoring

Finally, the keel bones were dissected from the hens and visually scored as to the gross presence or absence of fractures, deviations, and keel tip fractures. Keels were considered fractured if bony deposits were present along the ventral or lateral sides of the bone (indicating a fracture callus), and were considered deviated if the keel’s ventral aspect was not straight or the ventral ridge showed a divot. The cartilaginous aspects of the keel were not scored. As the presence of fractures, deviations, or keel tip fractures which were on the anatomical dorsum of the bone, or that were very slight, would not be palpatable, these were only counted when analyzing the accuracy of the radiography and sonography methods. Dissection scores were used to calculate the true prevalence of palpable abnormalities by removing the dorsal and slight damage.

### 2.5. Data Analysis

Intra-rater reliability was tested using a Cronbach’s Alpha test for each assessor. Tests were conducted for each category as well as all categories combined between scoring data for first and second round palpation. Inter-rater reliability was tested using a Fleiss’ Kappa test (an adaptation of Cohen’s Kappa for 3 or more assessors) for each category (fractures, deviation, tip abnormality) as a separate test. Tests were conducted on category data for the first-round palpations, second-round palpations, and combined category data. 

Inter-rater reliability was tested between second-round palpation scores of the 4 assessors, the second-round palpation scores of the 4 assessors and sonographs, as well as the second-round palpation scores of the 4 assessors and radiographs with Fleiss’ Kappa test for each category as above. 

Data for ultrasound and radiography scores were compared to dissection scores as the true prevalence of palpable keel bone abnormalities, and sensitivity (a true positive), specificity (a true negative), positive predictive value (PPV; ratio of true positives to combined true and false positives), and negative predictive value (NPV; ratio of true negatives to combined true and false negatives) were calculated. All analyses were performed using R (R 3.3.1 GUI 1.68 Mavericks build (7238)) with installed software packages “irr” and “psy” used for all statistical testing.

## 3. Results

The true prevalence and mean of the apparent prevalence of palpable keel bone fractures, deviations, and tip fractures are shown in Table 1. The average apparent palpable deviation prevalence among assessors (arithmetic mean of fracture prevalence between the two palpation blocks) varied from 43% (67.7% of true prevalence) to 52.5% (82.3% of true prevalence). The average apparent palpable fracture prevalence among assessors varied from 11% (44% of true prevalence of palpable abnormalities) to 24.2% (96.8% percent of true prevalence of palpable abnormalities). The average apparent palpable keel bone tip abnormality prevalence among assessors (determined as above) varied from 81.3% (92% of true prevalence) to 93% (105% of true prevalence). 

Alpha statistics calculated to interpret intra-rater reliability are shown in Table 2. Alpha Values represent the reliability between first and second-round palpation scores for each assessor by category of abnormality.

Kappa statistics calculated to interpret inter-rater reliability among the four assessors are shown in Table 3. Kappa values were obtained by Fleiss’ Kappa Test run among the four assessors for each category of palpable abnormality and each round of palpation, including rounds combined.

Using visual dissection bone scores as true prevalence of palpable keel bone abnormalities, sensitivity, specificity, positive predictive value (PPV), and negative predictive value (NPV), were calculated for sonography and radiography as shown in Table 4. Sensitivity, specificity, positive predictive value (PPV), and negative predictive value (NPV) were calculated for radiographical and sonographical detection of keel bone fractures, deviations, and tip fractures.

## 4. Discussion

### 4.1. Palpation Training

Assessors were able to correctly identify between 44% and 96.8% of keel bone fractures confirmed by dissection to be present in the palpated hens. Wilkins et al. [4] noted a much narrower range of accuracy (71–83%) in a palpation study, however, in their study, only three assessors were included and trained only to assess for old keel breakages displaying callus remodeling rather than including new fractures, as in this study. The wide range observed in the current study may reflect the range of palpation experience between assessors, as well as the improved detection scores of assessors after round 1 of palpation. Keel bone deviations were identified with lower percent agreement but higher precision than fractures (between 67.7% and 82.3% of confirmed prevalence), perhaps due to the presence of the pectoralis muscles inhibiting easy detection [6]. Detection of tip fractures was highest in both precision and accuracy, and reflected between 92–93% of true prevalence as determined at dissection. 

Agreement between assessors 1–4 in detecting any of the three types of keel injury was poor to moderate at best. Between palpation rounds, agreement only increased in the fracture category, and marginally decreased for deviations and tip fractures. This finding is supported by Buijs et al. [17] who noted that repeated palpation assessment might lead to error despite prior experience of assessors. Keel tip abnormality detection displayed the lowest reliability between assessors, despite being the category of highest accuracy and precision in detection. This may indicate that individual assessors were especially proficient at tip palpation, or that others were particularly inexperienced in this area. In either case, the low agreement between assessors, but high prevalence of tip injury, indicates that specialized training to detect such injuries is warranted when developing palpation training programs.

### 4.2. Radiography and Sonography

When compared to dissected and visually scored keel bones, radiography and sonography were both moderate to strong in sensitivity and specificity for the detection of keel bone injuries. Radiography’s high negative predictive value for fractures demonstrates its value as a rule-out tool for fracture injuries, as was expected. Radiography’s positive predictive value for tip fractures (100%), indicates that it is the radiologic tool of choice for detection of this type of injury. A recent study by Rufener et al. [18] developed a scoring system for assessing keel bone fractures from radiographs of laying hens with high intra-observer reliability, which further suggests the utility of radiography for detecting keel bone injuries. Sonography’s predictive values lagged somewhat behind those of radiography in all categories, however, its specificity for keel deviation detection was slightly higher than that of radiography.

Agreement between the techniques was excellent for the detection of fractures and tip fractures, but poor for keel deviations. This reflects the similarly poorer detection of keel deviations compared to fractures or tip fractures with either modality. While human assessors varied in accuracy of detection for keel deviations, all assessors showed comparable sensitivity for keel deviations, with the lowest-scoring assessor having a sensitivity of 57.6% (higher than that of sonography for deviations, and comparable to that of radiography). We speculate that due to the super-impositional nature of the image-capture in both radiography and sonography, keel deviations may be more difficult to detect with radiographic means than fractures or keel tip injuries without perfectly orthogonal placement of the sonographic probe or radiation beam in relation to the sagittal plane of the keel.

While sonography was somewhat less sensitive for keel bone injury detection, apart from tip fractures, where its sensitivity was 6% higher than radiography, the modality is advantageous in that it does not use harmful radiation. Its safety for operators, combined with its good injury detection rates, indicates that it may be a valuable tool for modified palpation training techniques in the future, similar to the one employed in this study. 

## 5. Conclusions

While radiographs present accessibility barriers to many training scenarios due to the need for specialized training, equipment, and radiation safety precautions, our results suggest a positive association between assessor learning and the ability to consult radiographs. Therefore, we feel that radiographs present a worthwhile learning advantage when available, particularly with the high reliability of scoring techniques for radiographs that have been developed [18]. We have identified that sonography may offer similar training benefits to assessors when performed by a trained operator, while eliminating radiation safety risks. As portable radiography becomes more accessible, its use with palpation will be important, as palpation alone likely underestimates the true prevalence of keel bone damage in a flock.

## Figures and Tables

**Table 1 animals-09-00894-t001:** The true prevalence of keel bone damage (% of total bones) as scored by keel bone dissection and the mean apparent prevalence on keel bone damage (% of total bones) as scored by palpation across four assessors.

	Deviation	Fracture	Tip Fracture
True Prevalence	25.5	64.7	90.2
Apparent Prevalence	48.0	17.6	88.8

**Table 2 animals-09-00894-t002:** Alpha values for the intra-reliability of four evaluators (and experience level) indicating agreement between the first and second rounds of palpation for keel bone deviations, fractures, and tip fractures.

	Experience	Deviation	Fracture	Tip Fracture
Evaluator 1	Moderate	0.576	0.899	0.374
Evaluator 2	High	0.643	0.846	0.918
Evaluator 3	High	0.788	0.893	0.745
Evaluator 4	None	0.587	0.657	0.885

**Table 3 animals-09-00894-t003:** Κappa values for the inter-reliability of four evaluators indicating agreement in the first, second, and combined rounds of palpation for keel bone deviations, fractures, and tip fractures.

	First-Round Palpation	Second-Round Palpation	Combined
Deviation	0.387	0.361	0.377
Fracture	0.528	0.552	0.540
Tip Fracture	0.124	0.118	0.121

**Table 4 animals-09-00894-t004:** The sensitivity (%), specificity (%), positive predictive value (PPV, %), and negative predictive value (NPV, %) of radiography and sonography in detecting keel bone deviations, fractures, and tip fractures.

Keel Damage	Technique	Sensitivity	Specificity	PPV	NPV
Deviations	Radiography	60.9	72.7	82.4	47.1
	Sonography	50.0	75.0	80.0	42.8
Fractures	Radiography	85.7	81.5	54.5	95.7
	Sonography	75.0	78.6	50.0	91.7
Tip Fracture	Radiography	84.4	100.0	100.0	28.5
	Sonography	90.9	67.0	96.8	67.0

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
