# Peer review of "The Reliability and Accuracy of Palpation, Radiography, and Sonography for the Detection of Keel Bone Damage"

_animals, 2019, doi:10.3390/ani9110894_

Round 1
Reviewer 1 Report
Keel bone damage is an important welfare problem in laying hens. In order to improve the welfare level, it is necessary to be able to detect this kind of damage in a good way. This paper may contribute to improving keel bone assessment. However, only 4 assessors is a small number.
Below are some suggestions to improve your manuscript.
L64-65: not all references (10-14) support both statements. Give the applicable references with the right statement.
Please insert empty lines above all subtitels, such as 2.1, 2.2, etc.
L93: does ‘over 100 palpations’ make somone ‘experienced? 100 is not so many and being experienced also depends on knowing what you feel, by adequate training and receiving feedback no wand then or recalibration.
L93: Please provide information about the degree of being expericend per assessor, also later in the results section.
L97: why ‘no more than twice’? Are you sure this is enough? Perhaps >2 gives a better impression.
2.3: can you, in the introduction maybe, give some brief information about the devices used, for example portable yes/no, size, specific environment needed yes/no, etc. Can the devices be used when visiting commercial farms?
L142-153: Please present these results in a table, which would make them more clear.
L159-160: Please provide information on the degree of experience of the different evaluators in the table. Providing such information makes your suggestions later in your article, that degree of experience might explain diffences between evaluaters, stronger.
L169: Please provide brief definitions of sensitivity, specificity, positive predictive value and negative predictive value.
L190: what do you mean with ‘precision’ and ‘accuracy’ in relation tot he earlier used sensitivity, specificity, ppv and npv?
L195: what do you mean with ‘prolonged palpations assessment’?
4.1 can be much stronger if in the results you mention the degree of experience per assessors.
L209: ‘similar detection levels’ with radiography?
L222: The results in table 3 support your remark about sensitivity, but the differences between radiography and sonography are very small, when talking about specificity and for deviations sonography has a higher sensitivity compared to radiography. Consider removing ‘and specific’.
L224: ‘harmful radiation’. Please provide some pros and cons like this one, for the different techniques in an overview in your introduction.
L230: ‘particularly when many assessors are trained at once’: why?
Please insert title ‘5 Conclusions’ where applicable.
Author Response
Keel bone damage is an important welfare problem in laying hens. In order to improve the welfare level, it is necessary to be able to detect this kind of damage in a good way. This paper may contribute to improving keel bone assessment. However, only 4 assessors is a small number.
The authors thank the reviewer for their helpful comments. While we agree that 4 assessors is a small number, we are confident that our results are fairly representative. We have spoked with several labs that work on keel bone damage, and they report similar results. This is of course personal communication, and untested statistically, but we hope our study will be the start of further investigations and show the importance of reporting this information in manuscripts on keel bone damage. Responses to the reviewer’s comments are below in bold.
Below are some suggestions to improve your manuscript.
L64-65: not all references (10-14) support both statements. Give the applicable references with the right statement.
Corrections have been made in the text.
Please insert empty lines above all subtitles, such as 2.1, 2.2, etc.
Line spaces have been inserted above all subtitles.
L93: does ‘over 100 palpations’ make someone ‘experienced? 100 is not so many and being experienced also depends on knowing what you feel, by adequate training and receiving feedback now and then or recalibration.
We came to this definition based on Petrik et al. (2013). They suggest an assessor train with 100 hens to gain a high accuracy with palpation and be considered experienced. We have added this to the paper.
L93: Please provide information about the degree of being experience per assessor, also later in the results section.
This information has been added to the Table.
L97: why ‘no more than twice’? Are you sure this is enough? Perhaps >2 gives a better impression.
We only palpated each keel twice, as we were trying to get the first impression of keel bone damage. From past work in our lab, we have found that assessors tend to second guess their initial impressions and become unsure with repeated palpations.
2.3: can you, in the introduction maybe, give some brief information about the devices used, for example portable yes/no, size, specific environment needed yes/no, etc. Can the devices be used when visiting commercial farms?
A paragraph on these technologies has been added to the Introduction. Some info has also been added to the Methods.
L142-153: Please present these results in a table, which would make them more clear.
A table with this data has been added.
L159-160: Please provide information on the degree of experience of the different evaluators in the table. Providing such information makes your suggestions later in your article, that degree of experience might explain differences between evaluators, stronger.
This information has been added.
L169: Please provide brief definitions of sensitivity, specificity, positive predictive value and negative predictive value.
Definitions have been added to the Methods.
L190: what do you mean with ‘precision’ and ‘accuracy’ in relation to the earlier used sensitivity, specificity, ppv and npv?
We realize that we used the term accuracy and precision in very general terms here. We have changed the term accuracy to % agreement (our intent for the term). Precision refers to PPV here, although we think the term precision works as they are used interchangeable and did not change this.
L195: what do you mean with ‘prolonged palpations assessment’?
This is referring to repeated palpations, the wording in the text has been changed to reflect this.
4.1 can be much stronger if in the results you mention the degree of experience per assessors.
This was added, and does show the general trend for more experienced assessors having better scores. Thank you for pointing this out.
L209: ‘similar detection levels’ with radiography?
This sentence has been clarified in the text.
L222: The results in table 3 support your remark about sensitivity, but the differences between radiography and sonography are very small, when talking about specificity and for deviations sonography has a higher sensitivity compared to radiography. Consider removing ‘and specific’.
This has been removed.
L224: ‘harmful radiation’. Please provide some pros and cons like this one, for the different techniques in an overview in your introduction.
Some of the pros and cons have been added to the Introduction.
L230: ‘particularly when many assessors are trained at once’: why?
This phrase has been clarified in the text.
Please insert title ‘5 Conclusions’ where applicable.
This has been inserted.

Reviewer 2 Report
I found your paper to be well written and interesting. The topic is very important since palpation is the preferred method in field investigation of keel bones, especially tip fractures may be difficult to detect. I really appreciate that you classified the keel bone damage as either deviation, fracture or tip fracture, this is a shortcoming in many KBF-papers.
Line 17-18: this sentence implies that KBF are strongly linked to alternative housing forms. The literature is not clear on this case. In addition, most KBF-estimates are based on palpation that, as you also point out, have a poor accuracy. Suggest you rephrase.
Line 46-49: the aim of your paper is not causes for keel bone fractures. Therefore, I suggest you delete this passage.
Line 50-55: This is an informative paragraph. It should be the start of your introduction.
Line 66: “to systems associated with higher keel fracture prevalence rates”. Hens in cages have less ability to move around. It could be speculated that less movement in the fracture site will create less callus and hence and underestimate of the fracture prevalence in hens from cages. Especially since the estimates are purely based on palpation. Therefore, I find it a bit alarming that you place all the blame on the housing system and imply that aviary systems are worse for hen welfare in terms of fractures.
Line 83: It would be interesting if you did the same study with hens from battery cages. Would this alter the results? Would the palpation accuracy be lower?
Line 90-: were all investigations of the keel performed after euthanasia?
Line 107-108: is this a stationary or portable X-ray?
Line 143: this is alarmingly high prevalences. How do you think this will be perceived the industry in the US?
Line 181: this is important information for future studies using palpation to detect KBF; palpation alone will likely underestimate the prevalence.
Line 199-201: this is a very important result for future studies and it may be the most interesting result from your study. suggest you lift this information to the abstract
Line 203-: Did radiography detect any fractures that were unmissed by visually scoring after dissection? Especially tip fractures can be difficult due to several fractures sites in the same area and very fine fracture lines.
Author Response
I found your paper to be well written and interesting. The topic is very important since palpation is the preferred method in field investigation of keel bones, especially tip fractures may be difficult to detect. I really appreciate that you classified the keel bone damage as either deviation, fracture or tip fracture, this is a shortcoming in many KBF-papers.
The authors thank the reviewer for their helpful comments. Responses are below in bold type.
Line 17-18: this sentence implies that KBF are strongly linked to alternative housing forms. The literature is not clear on this case. In addition, most KBF-estimates are based on palpation that, as you also point out, have a poor accuracy. Suggest you rephrase.
This line has been reworked in the text to address the reviewer’s concerns.
Line 46-49: the aim of your paper is not causes for keel bone fractures. Therefore, I suggest you delete this passage.
This paragraph has been deleted.
Line 50-55: This is an informative paragraph. It should be the start of your introduction.
This paragraph now begins the Introduction.
Line 66: “to systems associated with higher keel fracture prevalence rates”. Hens in cages have less ability to move around. It could be speculated that less movement in the fracture site will create less callus and hence and underestimate of the fracture prevalence in hens from cages. Especially since the estimates are purely based on palpation. Therefore, I find it a bit alarming that you place all the blame on the housing system and imply that aviary systems are worse for hen welfare in terms of fractures.
This sentence is not meant to place blame on any system. It simply reflects the trend in the literature that keel damage is more prevalent in alternative housing. We did include furnished cages in the sentences, we have added the word cages to make that more clear.
Line 83: It would be interesting if you did the same study with hens from battery cages. Would this alter the results? Would the palpation accuracy be lower?
We chose an enriched colony system as that was what was available (there were no conventionally housed hens at the facility). While this is all conjecture (without testing as suggested), we don’t think palpation accuracy would be affected per se by housing system. The literature suggests that hens in conventional cages have less keel bone damage, so it may have been harder to find. Having more uninjured keels may give the impression of higher accuracy and reliability, but it wouldn’t be measuring the ability of the techniques to detect keel bone damage.
Line 90-: were all investigations of the keel performed after euthanasia?
Yes, this has been clarified in the text.
Line 107-108: is this a stationary or portable X-ray?
This was portable, and this has been added to the text.
Line 143: this is alarmingly high prevalences. How do you think this will be perceived the industry in the US?
Yes, these are high. We did not make any comments on the prevalence purposefully. First, these are birds housed at a research facility, and not managed for production. Second, these birds are much older than would be found in US production. We felt there would be a high likelihood of keel bone damage in these hens, given the cage design and their age, which we felt would give some good variation for assessing the techniques. We were not trying to look at industry prevalence.
Line 181: this is important information for future studies using palpation to detect KBF; palpation alone will likely underestimate the prevalence.
Agreed, a statement pointing this out has been added to the Conclusion section.
Line 199-201: this is a very important result for future studies and it may be the most interesting result from your study. suggest you lift this information to the abstract
This has been added to the Abstract.
Line 203-: Did radiography detect any fractures that were unmissed by visually scoring after dissection? Especially tip fractures can be difficult due to several fractures sites in the same area and very fine fracture lines.
We only scored for presence/absence of fractures for all three techniques. However, we did not have any cases where a fracture was scored for the radiograph that was not also scored during visual dissection of the bone. Examining the number of fractures on the radiograph and comparing that to palpation scores would be an interesting next step.
Reviewer 3 Report
Please see my comments on the text marked by yellow color notes.

Author Response
We have carefully checked the highlighted areas.